# IMPELLA^®^ or Extracorporeal Membrane Oxygenation for Left Ventricular Dominant Refractory Cardiogenic Shock

**DOI:** 10.3390/jcm10040759

**Published:** 2021-02-14

**Authors:** Guillaume Schurtz, Natacha Rousse, Ouriel Saura, Vincent Balmette, Flavien Vincent, Nicolas Lamblin, Sina Porouchani, Basile Verdier, Etienne Puymirat, Emmanuel Robin, Eric Van Belle, André Vincentelli, Nadia Aissaoui, Cédric Delhaye, Clément Delmas, Alessandro Cosenza, Laurent Bonello, Francis Juthier, Mouhamed Djahoum Moussa, Gilles Lemesle

**Affiliations:** 1Cardiac Intensive Care Unit, Heart and Lung Institute, CHU Lille, 59000 Lille, France; guillaume.schurtz@hotmail.fr (G.S.); ouriel.saura@gmail.com (O.S.); vincent.balmette@gmail.com (V.B.); nicolas.lamblin@chru-lille.fr (N.L.); basile.verdier@chru-lille.fr (B.V.); 2Department of Interventional Cardiology for Coronary, Valves and Structural Heart Diseases, CHU Lille, Institut Coeur Poumon, Cardiology, 59000 Lille, France; flavienvincent@yahoo.fr (F.V.); sina.porouchani@gmail.com (S.P.); ericvanbelle@aol.com (E.V.B.); cedric.delhaye@chru-lille.fr (C.D.); alessandro.cosenza@hotmail.com (A.C.); 3Department of Cardiac Surgery, Institut Cœur Poumon, CHU Lille, INSERM U1011, Institut Pasteur de Lille, Université de Lille, 59000 Lille, France; natacha.rousse@chru-lille.fr (N.R.); emmanuel.robin@chru-lille.fr (E.R.); andre.vincentelli@chru-lille.fr (A.V.); francis.juthier@chru-lille.fr (F.J.); mouhamed.moussa@chru-lille.fr (M.D.M.); 4Heart and Lung Institute, University Hospital of Lille, Institut Pasteur of Lille, Inserm U1011 and FACT (French Alliance for Cardiovascular Trials), F-75000 Paris, France; 5Department of Cardiology, Hôpital Européen Georges Pompidou, Assistance Publique des Hôpitaux de Paris, 75015 Paris, France; etienne.puymirat@aphp.fr; 6Department of Critical Care Unit, Assistance Publique-Hôpitaux de Paris (AP-HP), Hôpital Européen Georges Pompidou (HEGP), Université Paris-Descartes, 75015 Paris, France; nadia.aissaoui@aphp.fr; 7INSERM UMR-1048, Intensive Cardiac Care Unit, Rangueil University Hospital, 31400 Toulouse, France; delmas.clement@chu-toulouse.fr; 8Cardiology Department, Mediterranean Association for Research and Studies in Cardiology (MARS Cardio), 13015 Marseille, France; laurent.bonello@ap-hm.fr; 9Centre for CardioVascular and Nutrition Research (C2VN), Aix-Marseille Univ, INSERM 1263, INRA 1260, Hopital Nord, 13015 Marseille, France

**Keywords:** IMPELLA^®^, extracorporeal membrane oxygenation, mechanical circulatory support, cardiogenic shock

## Abstract

Mechanical circulatory support (MCS) devices are effective tools in managing refractory cardiogenic shock (CS). Data comparing veno-arterial extracorporeal membrane oxygenation (VA-ECMO) and IMPELLA^®^ are however scarce. We aimed to assess outcomes of patients implanted with these two devices and eligible to both systems. From 2004 to 2020, we retrospectively analyzed 128 patients who underwent VA-ECMO or IMPELLA^®^ in our institution for refractory left ventricle (LV) dominant CS. All patients were eligible to both systems: 97 patients were first implanted with VA-ECMO and 31 with IMPELLA^®^. The primary endpoint was 30-day all-cause death. VA-ECMO patients were younger (52 vs. 59.4, *p* = 0.006) and had a higher lactate level at baseline than those in the IMPELLA^®^ group (6.84 vs. 3.03 mmol/L, *p* < 0.001). Duration of MCS was similar between groups (9.4 days vs. 6 days in the VA-ECMO and IMPELLA^®^ groups respectively, *p* = 0.077). In unadjusted analysis, no significant difference was observed between groups in 30-day mortality: 43.3% vs. 58.1% in the VA-ECMO and IMPELLA^®^ groups, respectively (*p* = 0.152). After adjustment, VA-ECMO was associated with a significant reduction in 30-day mortality (HR = 0.25, *p* = 0.004). A higher rate of MCS escalation was observed in the IMPELLA^®^ group: 32.3% vs. 10.3% (*p* = 0.003). In patients eligible to either VA-ECMO or IMPELLA^®^ for LV dominant refractory CS, VA-ECMO was associated with improved survival rate and a lower need for escalation.

## 1. Introduction

Despite improvements in heart failure (HF) pharmacological therapies and widespread use of timely reperfusion in acute myocardial infarction (MI), cardiogenic shock (CS) is still associated with a dismal prognosis and major resource utilization [1]. Standard of care and current guidelines support the use of vasoactive agents (inotropes and/or vasopressors) to maintain adequate cardiac output [2]. This strategy however exposes the patients to increased afterload, increased myocardial oxygen demand and higher risk of arrhythmias that could lead at the end to impaired perfusion, further clinical deterioration and death [3].

Mechanical circulatory support (MCS) devices could overcome these pitfalls and restore cardiac output, improve systemic hemodynamic and subsequently reduce mortality [4,5]. Despite the lack of robust evidences, use of these devices significantly rose in the past decade but may also be associated with potential life-threatening complications [6]. Therefore, a careful multimodal evaluation by a dedicated Shock Team (critical care physicians, cardiac intensivists, interventional cardiologists, cardio-thoracic surgeons, advanced heart failure specialists, and anesthesiologist) is recommended to ensure correct CS phenotyping, therefore allowing tailored and early interventions with adequate MCS selection.

Currently, in most tertiary European cardiac centers, veno-arterial extracorporeal membrane oxygenation (VA-ECMO) and IMPELLA^®^ are the most frequently devices implanted for refractory CS patients. Patients with a left ventricle (LV) dominant CS and without severe hypoxemia could be supported either with VA-ECMO or with IMPELLA^®^. Both have however different hemodynamic properties, with specific advantages/disadvantages and level of support. To date, direct comparison of these two devices is however limited and obviously highly challenging. Indeed, whether the choice would affect outcomes is poorly described.

Thus, we aimed in the present analysis to compare the characteristics and outcomes of patients implanted with VA-ECMO vs. IMPELLA^®^ for refractory LV dominant CS, depending on the first implanted device in a selected CS population eligible to both devices.

## 2. Materials and Methods

### 2.1. Population and Design

The present study is a single-center, retrospective, observational registry conducted at the Heart and Lung Institute, University Hospital of Lille, France. Between April 2004 and March 2020, all patients requiring MCS using either IMPELLA^®^ or VA-ECMO (irrespective of the first chosen device) for LV dominant refractory CS were included in the present registry (*n* = 262). All patients received standard CS therapies (adequate filling status, vasoactive drug use, and correction of the underlying cause) according to CS guidelines and local protocols.

The present analysis aimed to compare patients who were first implanted with the IMPELLA^®^ device (*n* = 31) to those first implanted with VA-ECMO (*n* = 97) as MCS device (Figure 1). For that purpose, we restrained the population to patients eligible to both devices (*n* = 128). Therefore, patients with severe hypoxemia (not related to acute heart failure), concomitant significant right ventricle (RV) failure, LV thrombus, and/or presenting with refractory cardiac arrest were excluded (*n* = 134).

For each specific case, a dedicated Shock Team was activated and discussed timing and choice of MCS insertion based on a local protocol of refractory cardiogenic shock management (Appendix A). Data were entered into a comprehensive database.

### 2.2. CS Definition

CS was defined by a combination of impaired hemodynamic from cardiac origin (systolic blood pressure (SBP) < 90 mmHg for > 30 min or use of catecholamines to maintain SBP > 90 mmHg; cardiac index < 1.8 L/min/m^2^ without support or < 2.2 L/min/m^2^ with support), pulmonary congestion and impaired end-organ perfusion (confusion, cold/clammy skin and extremities, urine output < 30 mL/h, or lactate > 2 mmol/L).

RV failure was defined by a TAPSE (Tricuspid Annular Plane Systolic Excursion) < 15 and/or S (tissular Doppler) < 9 cm/s on transthoracic echocardiography (TTE).

### 2.3. Devices Implantation and CS Management

Experienced physicians implanted all devices. VA-ECMO was inserted surgically or percutaneously when possible (stable condition and suitable vascular access). At first, VA-ECMO was central in 6 cases. All the remaining patients underwent peripheral VA-ECMO (*n* = 91).

A centrifugal pump was combined with an oxygenator and limb reperfusion was routinely implanted. VA-ECMO flow was adapted to residual cardiac function and macro/microcirculatory parameters (mean arterial pressure (MAP), lactate, and central venous oxygen saturation (ScvO2)), and for an optimal right ventricular unloading. Implantation was performed with the assistance of experienced perfusionists and cardio-thoracic surgeons at the bedside, in the catheterization laboratory or operating room depending on the context and patient’s hemodynamic conditions.

IMPELLA^®^ pumps were inserted percutaneously for CP devices (*n* = 26, 84%), or with an axillary surgical approach (arteriotomy and vascular graft) for 5.0 devices (*n* = 5, 16%). Choice was based on hemodynamic parameters, level of support needed and arterial access feasibility. All pumps were intended to run on the highest possible program level to achieve the best LV unloading and/or hemodynamic performance. Chest radiography and trans-thoracic echocardiography (TTE) was performed at implantation and repeated daily to ensure correct device position.

All patients without contraindications received unfractionated heparin to target an anti-Xa between 0.2 and 0.4 U/mL, modulated by pump flow and presence of bleeding or thrombotic complications. General medications including vasoactive drugs were in accordance to current CS recommendations.

### 2.4. Endpoints

The primary end-point was 30-day all-cause death. Secondary endpoints were 6-month all-cause mortality, 30-day and 6-month need for long-term LV assist device (LVAD), and/or heart transplantation. We also assessed a composite endpoint of all-cause death, LVAD implantation or heart transplant at 30 days and 6 months.

MCS escalation: In the IMPELLA^®^ group, refractory CS that required implantation of VA-ECMO on top of IMPELLA^®^ defined MCS failure because of persistent tissue hypoperfusion despite appropriate medical therapy and supposed from a cardiac origin. In the VA-ECMO group, MCS escalation was defined as insufficient LV unloading during VA-ECMO therapy leading to IMPELLA^®^ insertion, characterized by a combination of clinical (arterial pulsatility loss), radiological (pulmonary oedema), echocardiographic criteria (LV distension, severe mitral insufficiency, stasis and/or LV thrombosis), or an elevated pulmonary capillary wedge pressure despite usual measures. In addition, 1 patient was unloaded with fenestration and one with intra-aortic balloon pump (IABP) insertion. Although these 2 cases could be considered as MCS failure, they were not considered as MCS escalation in this analysis.

Cardiac recovery was defined by a durable improvement of myocardial parameters allowing definite MCS weaning without the need for LVAD or heart transplantation during follow-up.

Specific MCS devices adverse events were in accordance with a recent consensus statement endorsed by the MCS academic research consortium [7]. Finally, major bleeding events, stroke and sepsis were also evaluated at 6 months. Stroke and sepsis were defined in line with current definitions [8,9]. Bleeding events were based on the Bleeding Academic Research Consortium (BARC) standardized definitions and major bleeding was defined by a BARC > 2 [10].

### 2.5. Statistical Analysis

Qualitative categorical variables are presented as numbers (percentages) and were compared using Chi-2 or Fisher tests as appropriate. Quantitative continuous variables are presented as mean (±standard deviation (SD)) and compared using Student’s *t*-test. Survival analyses were performed using the Kaplan–Meier method and were compared by a log-rank test. For multivariable analysis, in the primary analysis (model 1), multivariate Cox analyses were used to determine the association between variables of interest (based on previous literature) and 30-day and 6-month mortalities after adjusting for age, sex, diabetes, acute myocardial infarction at admission, left ventricle ejection fraction (LVEF) at admission, lactate level, creatinine level, hemoglobin level at admission and VA-ECMO first group. A second model (model 2) was also performed using a different approach. In the model 2, all variables significantly (*p* < 0.05) different between the 2 groups (VA-ECMO vs. IMPELLA^®^) at 30 days and at 6 months were selected as covariables. For both models, hazard ratios (HRs) and 95% confidence intervals (CIs) were calculated. The proportional hazards assumption was tested and satisfied for all variables. Collinearity was excluded by constructing a correlation matrix between candidate predictors. A *p* value < 0.05 was considered as statistically significant. All statistical analyses were performed using SPSS 25.0 software (IBM Company, Armonk, New York, NY, USA).

The authors are solely responsible for the design of this study, all analyses, the editing of the paper and its final content. Our study complied with the “Declaration of Helsinki”.

## 3. Results

### 3.1. Baseline Characteristics

During the study period, 262 patients were implanted in our tertiary center (University hospital of Lille, France) with a MCS device for refractory CS. Among these, 128 patients were judged eligible to both MCS devices, either VA-ECMO or IMPELLA^®^ (Appendix A). VA-ECMO was used as first MCS device in 97 patients and IMPELLA^®^ in 31 patients (mostly IMPELLA-CP^®^, 84%).

Baseline characteristics of the population are summarized in Table 1. Mean age was 53.8 ± 13.1 year-old and 72.7% of the patients were male. Altogether, 19.5% of the patients had diabetes mellitus. Patients who underwent VA-ECMO as the first MCS device were significantly younger than those who underwent IMPELLA^®^: 52 ± 12.4 vs. 59.4 ± 13.8 (*p* = 0.006).

At admission, CS was mainly the consequence of acute myocardial infarction (56.3%). LVEF was significantly lower in patients who underwent VA-ECMO as the first MCS device: 19.1 ± 13.8 vs. 27.2 ± 17.6 (*p* = 0.017). Lactate level was higher: 6.84 ± 5.33 vs. 3.03 ± 1.57 mmol/L (*p* < 0.001) and hemoglobin level was lower 11.3 ± 2.4 vs. 13.5 ± 2.6 g/dL (*p* < 0.001) in VA-ECMO first patients. There was no difference for other baseline characteristics.

### 3.2. Outcomes

Duration of MCS was similar between both groups, respectively 9.4 ± 10.1 days in the VA-ECMO group and 6 ± 5 days in the IMPELLA^®^ group (*p* = 0.077).

Outcomes are summarized in Table 2. In our study, 30-day (primary endpoint) and 6-month mortality rates were 46.9% and 52.3%, respectively. The composite of all-cause death, LVAD or heart transplantation was of 68.8% at 30 days and 73.4% at 6 months. At 6 months, recovery was observed in 38 patients (29.7%).

In the univariate analysis, there was overall no significant difference between the VA-ECMO first group and IMPELLA^®^ first group in 30-day (primary endpoint) and 6-month mortality rates (43.3 % vs. 58.1%, *p* = 0.152 and 48.5% vs. 64.5%, *p* = 0.119) (Table 3 and Appendix A, Figure 2 and Figure 3). After adjustment (multivariate Cox analysis, model 1), VA-ECMO as the first MCS device implanted was significantly associated with better survival at 30 days and 6 months: HR = 0.25 (0.10–0.65) (*p* = 0.004) and 0.27 (0.11–0.64) (*p* = 0.001), respectively (Table 4 and Appendix A). Results were highly concordant with the model 2 (Appendix A). The composite of all-cause death, LVAD or heart transplantation was similar between both groups at 30 days and 6 months: 69.1% vs. 67.7%, *p* = 0.889 and 73.2% vs. 74.2%, *p* = 0.913, respectively. No difference was observed between both groups in other endpoints studied (Table 2) except for a lower rate of heart transplantation in the IMPELLA^®^ first group.

Of note, during follow-up, 15.6% of the patients (*n* = 20) required MCS escalation. The rate of MCS escalation was significantly lower in the VA-ECMO first group: 10.3% of the VA-ECMO first patients were unloaded with IMPELLA^®^ combination and 32.3% of the IMPELLA^®^ first patients were upgraded with VA-ECMO to ensure adequate perfusion (*p* = 0.003). In addition, MCS escalation was required earlier after first MCS implantation in the IMPELLA^®^ first group: 1.1 ± 0.9 vs. 5 ± 5 days (*p* = 0.026). Thirty-day and 6-month mortalities were higher in patients requiring MCS escalation during follow-up: 41.7% vs. 75% at 30 days and 47.2% vs. 80% at 6 months. Mortality was however similar between groups in those patients who required escalation: 70% vs. 80% (*p* = 0.999) at 30 days and 70% vs. 90% (*p* = 0.582) at 6 months in the IMPELLA^®^ and VA-ECMO group, respectively.

## 4. Discussion

In our selected population (patients eligible to both type of devices), IMPELLA^®^ devices were used in less severe patients as illustrated by a higher baseline LVEF and lower lactate level. All-cause 30-day and 6-month mortality rates were 46.9% and 52.3% respectively in the overall population and comparable with previous literature [11,12,13,14,15,16,17]. In the univariate analysis, VA-ECMO tends to be associated with better outcomes (not significant). Importantly, after adjustment, VA-ECMO first patients showed however significantly better 30-day and 6-month outcomes as compared to IMPELLA^®^ first patients (HR = 0.25 for 30-day mortality, *p* = 0.004). In addition, a 3-fold lower need for MCS escalation was observed in the VA-ECMO group.

To date, head-to-head comparison of IMPELLA^®^ versus VA-ECMO is limited since only a few studies focused on this topic in the literature [11,12,13,14,15,16,17]. In summary, all of these show similar outcomes between groups. Importantly, it should however be acknowledged that in these previous studies, heterogeneous patients were included with a high rate of patients with cardiac arrest (from 23% to 63%). By contrast, patients with cardiac arrest were excluded from our analysis since prognosis was rather related to neurological than cardiac disorders in this subset of patients. In addition, patients with right ventricle failure and/or severe hypoxemia were often not excluded in the previous literature. As a matter of fact, these previous studies did not select patients eligible to both strategies as we did and patient characteristics therefore largely differed between groups, even more than what we had observed in our analysis (biases difficult to account for even after adjustment or propensity score matching). In addition, the size of some of these studies did not allow statistical adjustment between groups.

An elegant recent animal study aimed to address hemodynamic properties of IMPELLA^®^ and VA-ECMO under similar hemodynamic conditions at equal flow rates (P8 for IMPELLA^®^ CP and pump flow at 3.2 L/min for VA-ECMO) in 12 swine females with induced acute myocardial infarction-CS [18]. At baseline, MAP and arterial lactate were similar in IMPELLA^®^ and VA-ECMO groups (31 vs. 34 mmHg, *p* = 0.47 and 3.8 vs. 3.0 mmol/L, *p* = 0.26 respectively). After 60 min of support, both devices restored lactate levels (1.7 mmol/L for IMPELLA^®^ group vs. 2.4 for VA-ECMO group, *p* = 0.75). Better unloading was achieved with IMPELLA^®^ (pressure volume area 2456 vs. 7101 mmHg.mL^−1^, *p* = 0.004) but a greater improvement in venous oxygen saturation was observed in the VA-ECMO group (venous O_2_ cerebral saturation 33 vs. 69%, *p* = 0.04). In this model, results highlight similar performance of IMPELLA^®^ and VA-ECMO to normalize tissular perfusion. Translation of these hemodynamic data into clinical practice and patient’s outcomes is however challenging: in this study, MCS was started immediately after CS initiation, a situation rarely encountered in a real-life setting.

MCS escalation should be regarded as the failure of the first chosen device. In our study, support upgrade was significantly more frequent in the IMPELLA^®^ first group. This point may be of main importance in daily practice since close to 1/3 of the patients in the IMPELLA^®^ first group required escalation (as compared to only 10% in the VA-ECMO group). Akanni et al. [17] reported their experience with MCS escalation, whether it was related to insufficient perfusion or refractory LV distension management. As expected, adding IMPELLA^®^ to VA-ECMO significantly reduced pulmonary artery pressures and adding VA-ECMO to on-going IMPELLA^®^ support significantly improved perfusion indices (lactate, ScvO2) and oxygenation (P/F ratio). In their prospective study, Garan et al. [14] noted that, overall, 47% of their patients were upgraded with almost a 3-fold higher rate in the IMPELLA^®^ first group (64.5% vs. 25%) similarly to our study. The question is: insufficient perfusion (reason for upgrade in the IMPELLA^®^ group) worse than the need for LV unloading (reason for upgrade in the VA-ECMO group) in terms of patient outcomes? The answer is not easy and observational data showed conflicting results. If VA-ECMO upgrade in IMPELLA^®^ patients were associated with better survival in the study of Mourad et al. (50% vs. 33% at discharge) [16], the opposite effect was observed in the study of Garan et al. [14] with a survival rate to discharge 75% of VA-ECMO first patients versus 50% for IMPELLA^®^ first patients. In our study, as expected, 30-day and 6-month mortalities were higher in upgraded patients; but this increase was not significantly influenced by the group (30-day mortality of 70% in VA-ECMO first vs. 80% in IMPELLA^®^ first patients, *p* = non-significant). Of note, we observed that time between first MCS implantation and need for escalation was significantly shorter for IMPELLA^®^ first patients (1.1 ± 0.9 days vs. 5 ± 5 days, *p* = 0.026). Timing of MCS upgrade was not available in previous studies on combined support since most of them mixed concomitant implantation of both devices, IMPELLA^®^ first/VA-ECMO staged, and VA-ECMO/IMPELLA^®^ staged [19]. Noteworthy, it is important to note that an intentional early combination of support devices (a situation very different from the need for MCS escalation) holds promise (despite a higher risk of complications), and recent studies suggest a potential mortality benefit of an early VA-ECMO and IMPELLA^®^ association [20,21] to fulfill essential conditions (adequate perfusion, LV unloading) for improving CS mortality.

Sepsis, major bleeding, vascular complication, or stroke are serious concerns that increase mortality in CS patients and could even compromise durable HF therapy candidacy (LVAD/heart transplantation) [22]. It is very difficult to reliably appreciate the safety profile of MCS because standard definitions were lacking before a recent expert consensus statement publication [7]. In our study, as compared to the most previous literature, rates of major complications were similar between both groups. In contrast to our results and those of Garan et al. [14] and Mourad et al. [16], Karami et al. [15] however observed more device-related vascular complications in the VA-ECMO group (39.5% vs. 16.7%, *p* = 0.001).

Our study (highly selected population) emphasizes the fact that after adjustment, VA-ECMO was associated with better outcome and a less frequent need for MCS escalation. Therefore, our results suggest that IMPELLA^®^ use (a quick, easy, and less invasive technique as compared to VA-ECMO) should be restricted to the less severe patients and at the very early phase of the CS depending on local facilities. Indeed, CS patients are initially often referred to PCI-capable hospitals but without advanced MCS capabilities. From a pathophysiological view, transvalvular devices are very attractive by solving the CS hemodynamic equation (systemic and coronary perfusion, ventricular and hepatorenal unloading) especially for AMI-CS patients without major organ failure or profound hypoperfusion. IMPELLA^®^ devices could find a real central place for preshock or mild CS management and probably before primary percutaneous coronary intervention (PCI) [23,24]. Current studies are underway to solve these major issues and better define the role and timing of MCS deployment for this subset of patients.

## 5. Strengths and Limitations

In our study, excluding RV failure, cardiac arrest, LV thrombus, or severe hypoxemia offers a homogeneous population regarding the CS phenotype with eligibility for either VA-ECMO or IMPELLA^®^. It therefore allows a fair comparison of both devices. However, the retrospective design has important limitations with inherent biases. This is a single center study that only reflects practices in our institution, which may not be generalizable worldwide. Finally and unfortunately, detailed hemodynamic data were not available for our patients, and MCS escalation effects cannot be precisely assessed.

## 6. Conclusions

In our study, as consistently observed in the literature in the field, VA-ECMO implanted patients for refractory CS showed more severe hemodynamic compromise as compared to IMPELLA^®^ patients. In a restricted population eligible for both devices and after adjustment, VA-ECMO was associated with improved survival rate and a lower need for MCS escalation. In advanced stages of CS, VA-ECMO should be preferred and promptly started to ensure adequate perfusion in critical settings. IMPELLA^®^ device, by its attractive properties, could be considered in less severe situations to promote myocardial recovery and reverse the downward spiral of CS. Randomized data are needed to confirm our findings.

## Figures and Tables

**Figure 1 jcm-10-00759-f001:**
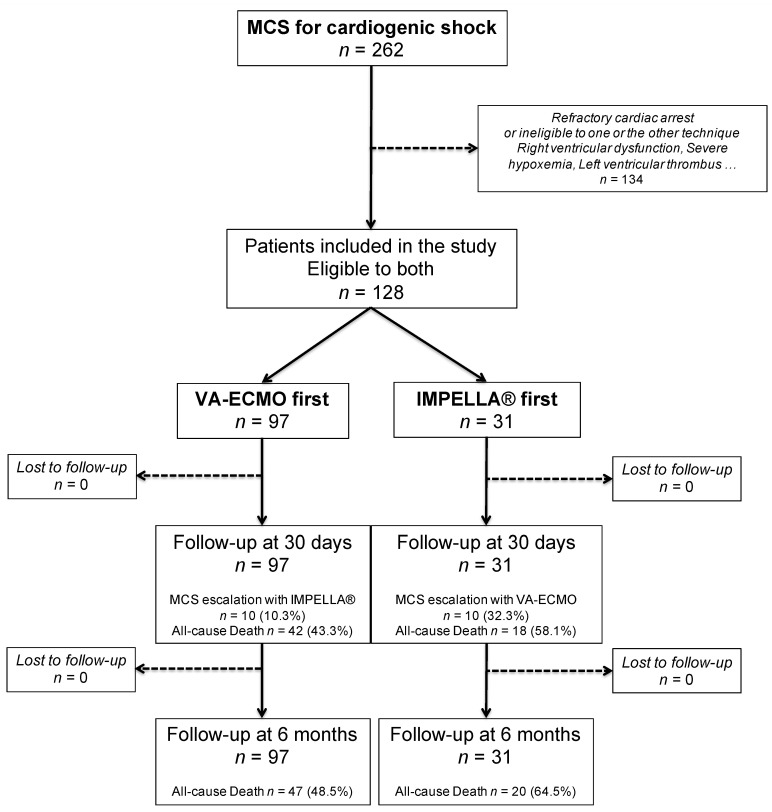
Flow chart of the study. MCS = mechanical circulatory support; VA-ECMO = veno-arterial extracorporeal membrane oxygenation; LV = left ventricle.

**Figure 2 jcm-10-00759-f002:**
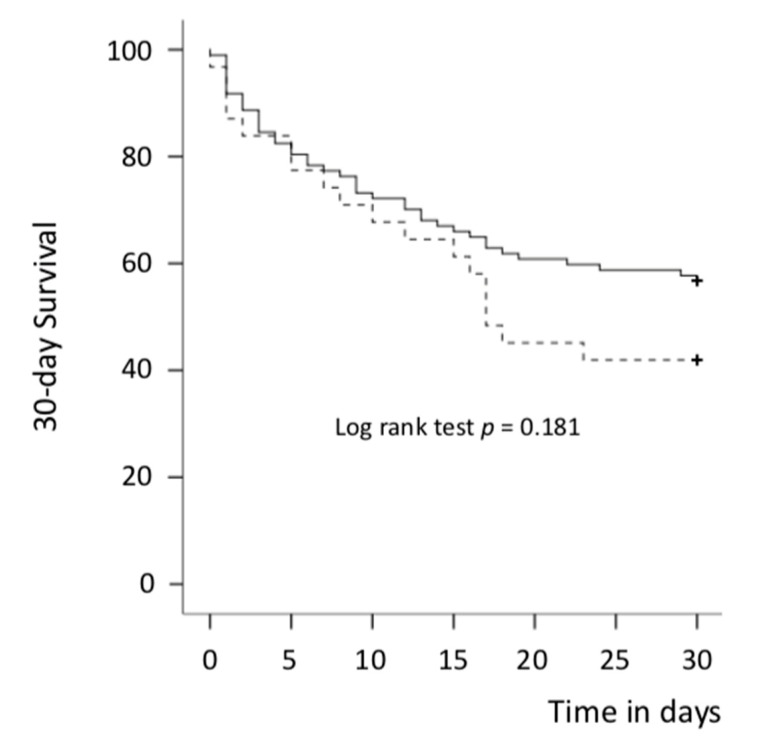
Thirty-day survival curves for all-cause death (Kaplan–Meier curves) in both groups. Solid line: VA-ECMO first group and dotted line: IMPELLA^®^ first group; VA-ECMO = veno-arterial extracorporeal membrane oxygenation. + = patient censored.

**Figure 3 jcm-10-00759-f003:**
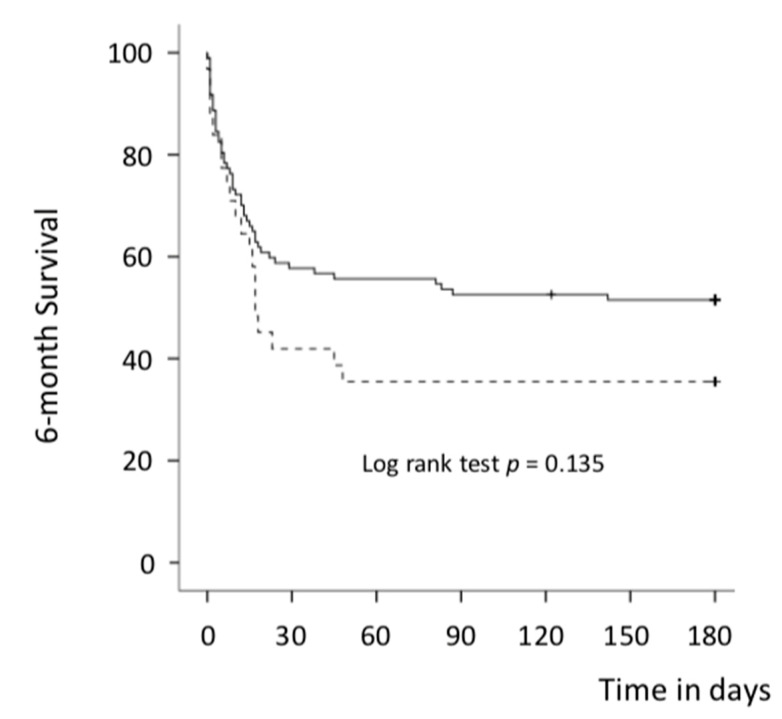
Six-month survival curves for all-cause death (Kaplan–Meier curves) in both groups. Solid line: VA-ECMO first group and dotted line: IMPELLA^®^ first group; VA-ECMO = veno-arterial extracorporeal membrane oxygenation; + = patient censored.

**Table 1 jcm-10-00759-t001:** Baseline characteristics of the population.

	Total Population *n* = 128	VA-ECMO First *n* = 97	IMPELLA^®^ First *n* = 31	*p*
Demographics and Medical History				
Age (years)	53.8 ± 13.1	52 ± 12.4	59.4 ± 13.8	0.006
Sex (male), *n* (%)	93 (72.7%)	70 (72.2%)	23 (74.2%)	0.825
Body mass index (Kg/m^2^)	27 ± 4.9	27 ± 5.1	26.7 ± 4.3	0.735
Diabetes, *n* (%)	25 (19.5%)	18 (18.6%)	7 (22.6%)	0.660
History of coronary artery disease, *n* (%)	34 (26.6%)	26 (26.8%)	8 (25.8%)	0.913
History of stroke, *n* (%)	4 (3.1%)	2 (2.1%)	2 (6.5%)	0.260
Peripheral artery disease, *n* (%)	3 (2.3%)	1 (1%)	2 (6.5%)	0.154
Renal failure before admission, *n* (%)	7 (5.5%)	5 (5.2%)	2 (6.5%)	0.999
Admission				
Acute myocardial infarction, *n* (%)	72 (56.3%)	51 (52.6%)	21 (67.7%)	0.138
Left ventricular ejection fraction (LVEF,%)	21.8 ± 15.5	19.1 ± 13.8	27.2 ± 17.6	0.017
Creatinine (mg/L)	18 ± 10.1	18.5 ± 10.3	16.5 ± 9.6	0.353
Hemoglobin (g/dL)	11.9 ± 2.6	11.3 ± 2.4	13.5 ± 2.6	<0.001
Lactate (mmol/L)	5.82 ± 4.93	6.84 ± 5.33	3.03 ± 1.57	<0.001
ASAT (IU/L)	460 ± 749	498.3 ± 835	348.8 ± 397.9	0.340
ALAT (IU/L)	269.5 ± 552.1	325.6 ± 627.5	104.8 ± 101.9	0.054
PT (%)	63.2 ± 31.2	61.7 ± 33.9	69 ± 17	0.322
Bilirubin (mg/L)	13.2 ± 14.5	14 ± 15.2	10.3 ± 11.7	0.289
CRP-us (mg/L)	68.7 ± 75.9	68.2 ± 71.7	70.3 ± 88.7	0.903
Mechanical support				
VA-ECMO, *n* (%)	107 (83.6%)	97 (100%)	10 (32.3%)	-
IMPELLA^®^, *n* (%)	41 (32%)	10 (10.3%)	31 (100%)	-
IMPELLA-CP^®^, *n* (%)	34 (26.6%)	7 (7.2%)	27 (87.1%)	-
VA-ECMO as first device, *n* (%)	97 (75.8%)	97 (100%)	-	-
IMPELLA^®^ as first device, *n* (%)	31 (24.2%)	-	31 (100%)	-
Duration of MCS (days)	8.6 ± 9.2	9.4 ± 10.1	6 ± 5	0.077

ASAT = aspartate aminotransferase; ALAT = alanine aminotransferase; PT = prothrombin time; CRP = C-reactive protein; VA-ECMO = veno-arterial membrane oxygenation; MCS = mechanical circulatory support.

**Table 2 jcm-10-00759-t002:** 30-day and 6-month outcomes.

	Total Population *n* = 128	VA-ECMO First *n* = 97	IMPELLA First *n* = 31	*p*
30-Day Outcomes				
30-day mortality, *n* (%)	60 (46.9%)	42 (43.3%)	18 (58.1%)	0.152
Bridge to left ventricular assist device (LVAD, *n* %)	18 (14.1%)	14 (14.4%)	4 (12.9%)	0.831
Bridge to transplantation, *n* (%)	14 (10.9%)	14 (14.4%)	0	0.025
Composite of all-cause death, LVAD, transplantation, *n* (%)	88 (68.8%)	67 (69.1%)	21 (67.7%)	0.889
Need for MCS escalation, *n* (%)	20 (15.6%)	10 (10.3%)	10 (32.3%)	0.003
Time between first MCS and escalation (Days)	3.1 ± 4	5 ± 5	1.1 ± 0.9	0.026
Stroke, *n* (%)	15 (11.7%)	13 (13.4%)	2 (6.5%)	0.295
Sepsis, *n* (%)	46 (35.9%)	39 (40.2%)	7 (22.6%)	0.075
Renal replacement therapy, *n* (%)	60 (46.9%)	43 (44.3%)	17 (54.8%)	0.307
Major bleeding, *n* (%)	48 (37.5%)	39 (40.2%)	9 (29%)	0.263
Vascular complication related to MCS insertion, *n* (%)	21 (16.5%)	15 (15.5%)	6 (19.4%)	0.627
Myocardial recovery, *n* (%)	37 (28.9%)	29 (29.9%)	8 (25.8%)	0.662
Six-Month Outcomes				
6-month mortality	67 (52.3%)	47 (48.5%)	20 (64.5%)	0.119
Bridge to LVAD, *n* (%)	20 (15.6%)	15 (15.5%)	5 (16.1%)	0.999
Bridge to transplantation, *n* (%)	16 (12.5%)	16 (16.5%)	0	0.016
Composite of all-cause death, LVAD, transplantation	94 (73.4%)	71 (73.2%)	23 (74.2%)	0.913
Need for MCS escalation, *n* (%)	20 (15.6%)	10 (10.3%)	10 (32.3%)	0.003
Time between first MCS and escalation (Days)	3.1 ± 4	5 ± 5	1.1 ± 0.9	0.026
Stroke, *n* (%)	16 (12.5%)	14 (14.4%)	2 (6.5%)	0.353
Sepsis, *n* (%)	53 (41.4%)	43 (44.3%)	10 (32.3%)	0.089
Renal replacement therapy, *n* (%)	63 (49.2%)	44 (45.4%)	19 (61.3%)	0.279
Major bleeding, *n* (%)	52 (40.6%)	40 (41.2%)	12 (38.7%)	0.771
Vascular complication related to MCS insertion, *n* (%)	21 (16.5%)	15 (15.5%)	6 (19.4%)	0.627
Myocardial recovery, *n* (%)	38 (29.7%)	30 (30.9%)	8 (25.8%)	0.587

MCS = mechanical circulatory support.

**Table 3 jcm-10-00759-t003:** Variables associated with 30-day mortality after adjustment (univariate analysis).

	Total Population *n* = 128	Alive at 30 Days *n* = 68	Dead at 30 Days *n* = 60	*p*
Demographics and Medical History				
Age (years)	53.8 ± 13.1	51 ± 13.6	56.9 ± 11.8	0.011
Sex (male), *n* (%)	93 (72.7%)	43 (63.2%)	50 (83.3%)	0.011
Body mass index (Kg/m^2^)	27 ± 4.9	26.6 ± 4.8	27.4 ± 5	0.404
Diabetes, *n* (%)	25 (19.5%)	10 (14.7%)	15 (25%)	0.118
History of coronary artery disease, *n* (%)	34 (26.6%)	18 (26.5%)	16 (26.7%)	0.980
History of stroke, *n* (%)	4 (3.1%)	0	4 (6.7%)	0.042
Peripheral artery disease, *n* (%)	3 (2.3%)	0	3 (5%)	0.089
Renal failure before admission, *n* (%)	7 (5.5%)	5 (7.4%)	2 (3.3%)	0.456
Admission				
Acute myocardial infarction, *n* (%)	72 (56.3%)	32 (47.1%)	40 (66.7%)	0.026
LVEF (%)	21.8 ± 15.5	20.8 ± 14	23.1± 17.4	0.472
Creatinine (mg/L)	18 ± 10.1	16.1± 8.3	20.2 ± 11.6	0.021
Hemoglobin (g/dL)	11.9 ± 2.6	11.7 ± 2.6	12.1 ± 2.6	0.503
Lactate (mmol/L)	5.82 ± 4.93	4.83 ± 3.48	7.09 ± 6.12	0.013
ASAT (IU/L)	460 ± 749	442.3 ± 878.9	482.7 ± 546.3	0.770
ALAT (IU/L)	269.5 ± 552.1	209.6 ± 484.6	340.1 ± 619.3	0.194
PT (%)	63.2 ± 31.2	67.9 ± 36.5	57.1 ± 21.4	0.070
Bilirubin (mg/L)	13.2 ± 14.5	11.2 ± 11.4	15.9 ± 17.7	0.098
CRP-us (mg/L)	68.7 ± 75.9	57.7 ± 83.9	83.9 ± 86.1	0.086
Mechanical Support				
VA-ECMO, *n* (%)	107 (83.6%)	57 (83.8%)	50 (83.3%)	0.940
IMPELLA^®^, *n* (%)	41 (32%)	16 (23.5%)	25 (41.7%)	0.028
IMPELLA-CP^®^, *n* (%)	34 (26.6%)	15 (22.1%)	19 (31.7%)	0.141
VA-ECMO as first device, *n* (%)	97 (75.8%)	55 (80.9%)	42 (70%)	0.152
IMPELLA^®^ as first device, *n* (%)	31 (24.2%)	13 (19.1%)	18 (30%)	0.152
Duration of MCS (Days)	8.6 ± 9.2	9.5 ± 11.1	7.5 ± 6.4	0.230

Abbreviations = cf. Table 1.

**Table 4 jcm-10-00759-t004:** Variables associated with 30-day mortality after adjustment (Cox multivariate analysis, model 1).

	Hazard Ratio	95% Confidence Interval	*p*
Age (per year)	1.03	0.99–1.07	0.073
Sex male	5.70	1.85–17.51	0.002
Diabetes	1.51	0.68–3.56	0.309
Acute myocardial infarction at admission	2.79	1.29–6.03	0.009
LVEF (per %)	0.99	0.97–.01	0.349
Lactate level (per one unit)	1.17	1.10–1.25	<0.001
Creatinine level (per one unit)	1.04	1.01–1.08	0.026
Hemoglobin level (per one unit)	0.92	0.81–1.05	0.236
VA-ECMO first	0.25	0.10–.65	0.004

Cox model: age, sex, diabetes, acute myocardial infarction at admission, left ventricle ejection fraction, lactate level, creatinine level, hemoglobin level, VA-ECMO first; variables of interest were selected based on the previous literature. LVEF = left ventricular ejection fraction; VA-ECMO = veno-arterial membrane oxygenation.

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
