# Peer review of "IMPELLA® or Extracorporeal Membrane Oxygenation for Left Ventricular Dominant Refractory Cardiogenic Shock"

_jcm, 2021, doi:10.3390/jcm10040759_

Round 1
Reviewer 1 Report
The authors performed a single-center retrospective observational study in 97 pts with VA-ECMO and 31 with Impella for treatment of cardiogenic shock and found that patients primarily treated with VA-ECMo had a better chance of survival (after adjustment of baseline characteristics).
Although this should be confirmed by prospective studies before any sound conclusions can be made, the findings are of potential interest.
Some things are not clear to me.
- which Impella is used? Percutaneous 2.5 or 3.5L or the surgical 5.0L? Changes in outcome?
- is duration of MCS corrected for death, e.g. is 30 day free MCS not a better measurement?
- how were variables used for adjustment chosen?
- frequency of LV unloading with Impella in case of VA-ECMO was only 10%. Was Impella first choice or only after IABP or fenestration. How many VA-ECMO were central/peripherel and effect on LV overload?
- Why use Cox regression with a fixed endoint or 30 days and 60 days
- there is a huge effect of adjusment. Crude OR (or HR) is not provided (in KM curve no difference) bur RR is 0.75 which change to 0.25 after adjustment! Which variabel has the largest effect on risk ratio?
- discussion is quit long. First paragraph can be replaced by major findings
Author Response
Reviewer #1
The authors performed a single-center retrospective observational study in 97 pts with VA-ECMO and 31 with Impella for treatment of cardiogenic shock and found that patients primarily treated with VA-ECMO had a better chance of survival (after adjustment of baseline characteristics).
Although this should be confirmed by prospective studies before any sound conclusions can be made, the findings are of potential interest.
Thank you very much for your interest in our study.
Some things are not clear to me.
- which Impella is used? Percutaneous 2.5 or 3.5L or the surgical 5.0L? Changes in outcome?
Thank you for your comment.
As stated in the manuscript (Table 1 and “results” section Page #10,12), IMPELLA 3.5L (or also known as CP) was used in the vast majority (84%) of the cases.
No IMPELLA 2.5 were implanted during this period in patients with cardiogenic shock. The use of IMPELLA 2.5 was restricted to patients with high-risk PCI in our center.
The remaining patients were implanted with the surgical IMPELLA 5.0L (16% of the cases).
The manuscript has been modified to clarify this point in the method section (Page #7).
- is duration of MCS corrected for death, e.g. is 30 day free MCS not a better measurement?
We agree that such an endpoint is of interest.
As stated in the method section (Page #9), “Cardiac recovery was defined by a durable improvement of myocardial parameters allowing definite MCS weaning without the need for LVAD or heart transplantation during follow-up.”
To our view, the definition we used for cardiac recovery is highly similar to the endpoint named here “30-day survival free from MCS”.
Cardiac recovery was assessed as a secondary endpoint in our manuscript (Table 2) and no difference was observed between groups at 30 days and at 6 months.
- how were variables used for adjustment chosen?
Selected variables for adjustment were chosen based on the previous literature. This is now stated in the method section (Page #9). We selected variables that were often associated with mortality in the previous literature in the field of cardiogenic shock.
This information was also added in the footnote of new Table 4 and Supplemental Table S2.
Please see also the answer to comment #1 and #2 of reviewer #2.
- frequency of LV unloading with Impella in case of VA-ECMO was only 10%. Was Impella first choice or only after IABP or fenestration. How many VA-ECMO were central/peripherel and effect on LV overload?
IMPELLA was the first choice in all cases to unload VA-ECMO patients who required it during follow-up. IABP was used in 1 patient and fenestration in 1 another patient. Although fenestration could be considered as MCS failure, it cannot be considered as MCS escalation by itself, which was our endpoint.
The manuscript has been modified to address this comment (method section, Page #8) and to clarify this point.
At first, VA-ECMO was central in 6 cases. All the remaining patients underwent peripheral VA-ECMO (n=91).
The manuscript has been modified to address this comment (method section, Page 7).
- Why use Cox regression with a fixed endoint or 30 days and 6 months
Although we had fixed endpoints at 30 days and 6 months, time of event occurrence during follow-up is important. To our view, it is absolutely not the same thing to die at day 1 or at day 29 in such very high-risk patients who often wait for either recovery, definite MCS by LVAD and/or heart transplantation.
Cox regression is the best way to take time of event occurrence into account. This the reason why we choose it.
This is also the reason why we display KM curves in Figures 2 and 3.
- there is a huge effect of adjusment. Crude OR (or HR) is not provided (in KM curve no difference) bur RR is 0.75 which change to 0.25 after adjustment! Which variabel has the largest effect on risk ratio?
As stated in our manuscript, VA-ECMO was implanted in more severe patients with a much higher degree of hemodynamic impairment as shown by creatinine and lactate level at admission.
Lactate level was the variables with the largest impact on the risk ratio as shown in the new Table 4 (RR of 17% per 1 mmol/L increase, which represents a huge effect).
Lactate was twice higher in patients with VA-ECMO as shown in Table 1.
These facts explain our results.
- discussion is quit long. First paragraph can be replaced by major findings
As requested, the discussion has been modified to shorten it. The first paragraph has been totally removed and the previous second paragraph, which focused on major findings of our study is now the first paragraph (Pages #19-20).
Reviewer 2 Report
The authors of the manuscript entitled "IMPELLA® or extracorporeal membrane oxygenation for left ventricular dominant refractory cardiogenic shock" show the results of a retrospective analysis of the results of 2 different MCS systems in those patients who could be treated with either of them. The main conclusion points out better results for VA-ECMO, based on the results of multivariate analysis.
As far as the interest of the topic is really high, I feel that the main conclusion is somewhat speculative and it should be supported by extended analysis:
- First, why performing the multivariate analysis on predetermined variables and not detailing the influence of them in the CURRENT set of observations? It would be helpful to show the univariate correlation of the selected variables with death rate in the studied group. This could lead to make a different multivariate model with totally different conclusions.
- Second, selecting 9 variables to perform a multivariate analysis on 60 events could be considered excessive. This emphasizes the need of results of the individual variables in this group (1st Point).
- Third, there is no difference in death + LVAD + heart transplant rates at 30 days / 6 months. Why picking up only the death rate and not the (very hard) combined endpoint to assess the efficacy? Almost 25% of the patients in the VA-ECMO group received LVAD / transplant vs. less than 10% in the Impella group.
- Fourth, very long inclusion period. Any differences in the systems use / preference along this time span? May it have influenced the results?
- Fifth, escalation implies the addition of the other system to first system used. Do we have information on the use of aortic counterpulsation balloon pump to unload LV in the VA-ECMO group? Should we consider it also an "escalation"?
In summary, I think performing the suggested additional statistical analyses could improve the message. Also, in my opinion, the combined variable death + LVAD + transplant better fits to offer a global message. At least, it should be detailed in the main conclusion.
Author Response
Reviewer #2
The authors of the manuscript entitled "IMPELLA® or extracorporeal membrane oxygenation for left ventricular dominant refractory cardiogenic shock" show the results of a retrospective analysis of the results of 2 different MCS systems in those patients who could be treated with either of them. The main conclusion points out better results for VA-ECMO, based on the results of multivariate analysis.
As far as the interest of the topic is really high, I feel that the main conclusion is somewhat speculative and it should be supported by extended analysis
Thank you for your interest in our study.
- First, why performing the multivariate analysis on predetermined variables and not detailing the influence of them in the CURRENT set of observations? It would be helpful to show the univariate correlation of the selected variables with death rate in the studied group. This could lead to make a different multivariate model with totally different conclusions.
- Second, selecting 9 variables to perform a multivariate analysis on 60 events could be considered excessive. This emphasizes the need of results of the individual variables in this group (1st Point).
Thank you for these relevant comments.
Statisticians usually recommend to perform a multivariate analysis using as co-variables those strongly associated with the outcome of interest in the previous literature rather than those associated with the outcome of interest in a single dedicated study. They also recommend to exclude variables with high collinearity. This is the reason why we made the choice to perform the analysis with predetermined variables (based on the previous literature). Age, sex, diabetes, lactate, creatinine, hemoglobin, LVEF, and acute MI at admission (n=8) have indeed all been strongly associated with mortality in the literature in patients with cardiogenic shock. In addition, as stated in the method section (Page #9) “Colinearity was excluded by constructing a correlation matrix between candidate predictors.” Other variables such as liver biomarkers for example (correlated with lactate level) were therefore not considered in our model.
In addition, this technique allowed us to use similar multivariate models for 30-day and 6-month mortality in order to make the manuscript easier to follow by readers.
We agree that to provide a univariate correlation of variables (as assessed in our study) with death will improve the manuscript. We have therefore added 2 new Tables (new Table 3 and new Supplemental Table S1) to address this specific point (as a consequence, previous Table 3 and Supplemental Table S1 are now new Table 4 and Supplemental Table S2).
Of note, at 30 days, variables significantly (p value <0.05) associated with death were: age, sex, history of stroke, acute MI at admission, creatinine and lactate (n=6 variables). At 6 months, variables associated with death were: age, sex, creatinine and lactate (n=4).
We also agree that 9 variables may be considered a bit excessive to perform a multivariate analysis on 60 events at 30 days and 67 events at 6 months.
As suggested by reviewer #2, we have performed new multivariate models at 30 days and 6 months including only variables significantly associated with the outcome of interest in our own dataset (excluding history of stroke since it concerned only 4 patients and is therefore not pertinent in a multivariate model).
These analyses are provided below. As you can see, these new models do not change the results overall. If editor agrees, we therefore suggest to leave our initial multivariate model unchanged to keep the manuscript simple and easy to follow by readers and to be in agreement with statisticians. We are however ready to provide these new models as supplemental materials or to replace our initial model by these new models if you feel it critical.
Variables associated with 30-day mortality after adjustment (Cox multivariate analysis, model with 6 variables)
|
|
Hazard ratio |
95% confidence interval |
p |
|
Age (per year) |
1.04 |
0.99-1.07 |
0.059 |
|
Sex male |
4.73 |
1.63-13.73 |
0.004 |
|
Acute myocardial infarction at admission |
3.22 |
1.44-7.16 |
0.004 |
|
Lactate level (per one unit) |
1.16 |
1.09-1.23 |
<0.001 |
|
Creatinine level (per one unit) |
1.05 |
1.01-1.10 |
0.012 |
|
VA-ECMO first |
0.38 |
0.18-0.74 |
0.007 |
Variables associated with 6-month mortality after adjustment (Cox multivariate analysis, model with 5 variables)
|
|
Hazard ratio |
95% confidence interval |
p |
|
Age (per year) |
1.04 |
1.01-1.07 |
0.011 |
|
Sex male |
3.43 |
1.59-7.41 |
0.002 |
|
Lactate level (per one unit) |
1.16 |
1.10-1.22 |
<0.001 |
|
Creatinine level (per one unit) |
1.04 |
1.01-1.07 |
0.003 |
|
VA-ECMO first |
0.34 |
0.17-0.67 |
0.002 |
- Third, there is no difference in death + LVAD + heart transplant rates at 30 days / 6 months. Why picking up only the death rate and not the (very hard) combined endpoint to assess the efficacy? Almost 25% of the patients in the VA-ECMO group received LVAD / transplant vs. less than 10% in the Impella group.
Thank you for your comment.
We however partially disagree on this specific comment.
In facts, to our view, there is no “harder” endpoint than total all-cause death.
MCS has never been a causal treatment in patients with cardiogenic shock. It will never treat the cause of the shock by itself. The goal of MCS is to wait for either cardiac recovery in some cases (after causal treatment: revascularization …), or long-term LVAD insertion and/or heart transplantation in some other cases. MCS should be considered as a transient solution to bear hemodynamic and cardiac function the time required.
Therefore, we would like to keep this combined endpoint as a secondary endpoint since it makes no real clinical sense by itself to us.
The goal of the present study is really to assess whether a device is superior to another to wait for recovery or definitive solution in such patients.
- Fourth, very long inclusion period. Any differences in the systems use / preference along this time span? May it have influenced the results?
Thank you for this comment. We agree that the period of inclusion is long but it allows us to include more patients and to have enough power for the statistical analyses.
The first IMPELLA implanted in our center was in 2013. Before this date, 27 patients had VA-ECMO in the present analysis, which represents a small number overall. Thereafter, systems and drugs were used using the same protocol over time as shown in Supplemental Figure S1.
If the very long inclusion period would have influenced the results, it would therefore have favor the IMPELLA device and decrease the benefit we observed in the present analysis of the VA-ECMO over the IMPELLA device (since IMPELLA appears after VA-ECMO in our center).
It is therefore unlikely that this point may have impacted our conclusion.
- Fifth, escalation implies the addition of the other system to first system used. Do we have information on the use of aortic counterpulsation balloon pump to unload LV in the VA-ECMO group? Should we consider it also an "escalation"?
IMPELLA was the first choice in all cases to unload VA-ECMO patients who required it during follow-up. IABP was used in 1 patient and fenestration in 1 another patient. Although fenestration could be considered as MCS failure, it cannot be considered as MCS escalation by itself, which was our endpoint.
The manuscript has been modified to address this comment (method section, Page #8) and to clarify this point.
In summary, I think performing the suggested additional statistical analyses could improve the message. Also, in my opinion, the combined variable death + LVAD + transplant better fits to offer a global message. At least, it should be detailed in the main conclusion.
Thank you again for your interest in our study.
We partially disagree on the point dealing with the combined endpoint as we explain it and we would like to keep it as a secondary endpoint (see answer to comment #3). As requested, we have added the suggested additional statistical analyses, which significantly improve the manuscript.
Reviewer 3 Report
Overall, the authors present a study that addresses an important question in the treatment of patients suffering from cardiogenic shock. Though the publication is generally well-written, the results as well as the discussion section are in parts hard to follow for the reader. This could be improved, if the following issues were addressed:
- In the results section too much redundant information, especially numbers and p-values are given in the text, that are already presented in the tables. This makes the text unnecessarily hard to follow. Considering the number of comparisons made for the present study a presentation in tables seems reasonable. The authors should limit the redundantly given numerical results to the key messages of the study. The rest should be the mere statements backed by the corresponding table.
- Overall, the discussion shows a good logical structure, but it needs to be more concise. Especially the third and fifth paragraph should be shortened.
- Undoubtedly, it is important to put one’s own conclusions into the context of existing literature. The authors provide a very thorough review of the literature with detailed citation of results and even p-values of other publications in their discussion section. This takes a lot of the focus of the discussion of their own valuable results. At the moment, the important message of the paper is mainly visible to the reader in the conclusion. This imbalance should be corrected before publication.
Author Response
Reviewer #3
Overall, the authors present a study that addresses an important question in the treatment of patients suffering from cardiogenic shock. Though the publication is generally well-written, the results as well as the discussion section are in parts hard to follow for the reader.
Thank for your interest in our study.
This could be improved, if the following issues were addressed:
- In the results section too much redundant information, especially numbers and p-values are given in the text, that are already presented in the tables. This makes the text unnecessarily hard to follow. Considering the number of comparisons made for the present study a presentation in tables seems reasonable. The authors should limit the redundantly given numerical results to the key messages of the study. The rest should be the mere statements backed by the corresponding table.
As requested, the result section has been shortened to avoid redundances with Tables.
- Overall, the discussion shows a good logical structure, but it needs to be more concise. Especially the third and fifth paragraph should be shortened.
- Undoubtedly, it is important to put one’s own conclusions into the context of existing literature. The authors provide a very thorough review of the literature with detailed citation of results and even p-values of other publications in their discussion section. This takes a lot of the focus of the discussion of their own valuable results. At the moment, the important message of the paper is mainly visible to the reader in the conclusion. This imbalance should be corrected before publication.
As requested, the discussion has been shortened to simplify messages and to focus on the key messages. All detailed information on the previous literature especially in 3rd and 5th paragraph have been removed and summarized in few sentences.
Round 2
Reviewer 1 Report
The authors have adequately respond to my questions. One minor thing is the respons on wheather to use Cox regression or not: "To our view, it is absolutely not the same thing to die at day 1 or at day 29....."
In this case I feel it is worse to dy at day 29 then on day first and Cox regression is not appropriate, but I don not feel the results would change significantly with anothe approach (log regression)
Author Response
Reviewer #1
The authors have adequately respond to my questions.
Thank you again for your interest in our study.
One minor thing is the respons on wheather to use Cox regression or not: "To our view, it is absolutely not the same thing to die at day 1 or at day 29..... »
In this case I feel it is worse to dy at day 29 then on day first and Cox regression is not appropriate, but I don not feel the results would change significantly with anothe approach (log regression)
Thank you for your comment.
When time of event occurrence matters as in the present analysis (as you also agree here and whatever which is felt the worse: early vs. late death), it is well recognized by statisticians that a Cox regression is the best way to take cofounders and time of event occurrence into account within the same analysis.
In addition, we also agree that results would not be significantly impacted by another approach.
Therefore, if editors agree we would like to keep the cox regression analysis in the revised version.
Reviewer 2 Report
Thanks for your response and additional work, that can improve the robustness of your work. I would like to ask for a few minor changes:
- ABSOLUTELY agree (of course): nothing can be compared to death as hard endpoint. But LVAD / transplant may be deemed as "organ death". Although you are right when arguing that being able to reach one of these therapies is not bad, everything can be discussed. My suggestion: just change the final sentence of your conclusion to VA-ECMO was associated with improved SURVIVAL RATE and … Because VA-ECMO was not associated with an improvement in your secondary endpoint.
- Very good job with the alternative multivariate model. What about detailing that you have performed it in both ways and referring the readers to the supplemental material?
Author Response
Reviewer #2
Thanks for your response and additional work, that can improve the robustness of your work.
Thank you again for your interest in our study.
I would like to ask for a few minor changes:
- ABSOLUTELY agree (of course): nothing can be compared to death as hard endpoint. But LVAD / transplant may be deemed as "organ death". Although you are right when arguing that being able to reach one of these therapies is not bad, everything can be discussed. My suggestion: just change the final sentence of your conclusion to “VA-ECMO was associated with improved SURVIVAL RATE and …” Because VA-ECMO was not associated with an improvement in your secondary endpoint.
The conclusion has been modified as requested in the abstract (Page #3) and in the main manuscript (Page #23).
- Very good job with the alternative multivariate model. What about detailing that you have performed it in both ways and referring the readers to the supplemental material?
As requested, we have added the results of this model #2 in supplemental materials (new Tables S3 and S4).
The statistical analysis paragraph (Page #9) and results section (Page 12) have been modified accordingly.
Reviewer 3 Report
The authors have implemented all suggestions sufficiently. The key message of the paper is now clearly transported.
Author Response
Reviewer #3
The authors have implemented all suggestions sufficiently. The key message of the paper is now clearly transported
Thank you again for your interest in our study.